# One-Step Genetic Modification by Embryonic Doral Aorta Injection of Adenoviral CRISPR/Cas9 Vector in Chicken

**DOI:** 10.3390/ijms25168692

**Published:** 2024-08-09

**Authors:** Chao Qin, Shengyao Jiang, Ke Xu, Jianshen Zhu, Liyuan Wang, Wenhao Yang, Fuquan Xiao, Kaixuan Yang, Qizhong Huang, He Meng

**Affiliations:** 1Shanghai Key Laboratory of Veterinary Biotechnology, Department of Animal Science, School of Agriculture and Biology, Shanghai Jiaotong University, Shanghai 200240, China; qin_chao@sjtu.edu.cn (C.Q.); jiangshengyao@sjtu.edu.cn (S.J.); keristina@sjtu.edu.cn (K.X.); zhujianshen@sjtu.edu.cn (J.Z.); kkkly88@sjtu.edu.cn (L.W.); yangwenhao@sjtu.edu.cn (W.Y.); xiaofuquan@sjtu.edu.cn (F.X.); 2Animal Husbandry and Veterinary Research Institute, Shanghai Academy of Agricultural Science, Shanghai 200030, China; yangkaixuan007@hotmail.com (K.Y.); huangqizh@163.com (Q.H.)

**Keywords:** chicken, CRISPR/Cas9, dorsal aorta, *KRT75L4*, adenovirus

## Abstract

In the avian species, genetic modification by cell nuclear transfer is infeasible due to its unique reproductive system. The in vitro primordial germ cell modification approach is difficult and cumbersome, although it is the main method of genetic modification in chickens. In the present study, the adenoviral CRISPR/Cas9 vector was directly microinjected into the dorsal aorta of chicken embryos to achieve in vivo genetic modification. The results demonstrated that *keratin 75-like 4* (*KRT75L4*), a candidate gene crucial for feather development, was widely knocked out, and an 8bp deletion was the predominant mutation that occurred in multiple tissues in chimeras, particularly in the gonad (2.63–11.57%). As we expected, significant modification was detected in the sperm of G0 (0.16–4.85%), confirming the potential to generate homozygous chickens and establishing this vector as a simple and effective method for genetic modification in avian species.

## 1. Introduction

The chicken provides large amounts of protein for the human diet as an important agricultural animal [1], and it is also an excellent biological research model organism for biology research [2]. In mammals, somatic cell nuclear transfer (SCNT) [3,4] has been successfully used to generate genetically modified species because of the easy accessibility of a single oocyte. In chickens, the efficient strategy used in mammals does not apply due to their unique reproductive system. After ovulation, an oocyte containing delicate yolk is transported into the oviduct and fertilized within the infundibulum. The resulting single zygote undergoes rapid proliferation, reaching approximately 55,000 cells during the 24 h period of egg laying [5,6], which makes it challenging to obtain a single zygote for genome modification in vitro [7,8]. In addition, obtaining a single oocyte attached to the yolk through surgery and transplanting it back after genetic modification is extremely difficult and uneconomical [9].

Primordial germ cells (PGCs) are the initial germ cell population formed during development, serving as precursors for both oocytes and spermatogonia. In mammalian species, PGCs migrate from the yolk sac to the genital ridge via mesentery. In chicken, PGCs migrate from the primitive streak and endoderm to the developing gonad through extraembryonic blood vessels [10], facilitating the genetic modification of PGCs. Over recent decades, advancements in PGC culture and gene editing technology have led to the widespread adoption of in vitro PGC genetic modification methods, which require the isolation and culture of PGCs, genome modification and screening of PGCs in vitro, and transplantation of genetically modified PGCs into the blood vessels [11], the subgerminal cavity [12] of recipient embryos, or testes of adult roosters [13]. Although the in vitro PGC modification approach is now well-established and has been wildly utilized in many avian species, the procedures are still technically difficult and cumbersome, and endogenous PGCs may compete with genetically modified PGCs after transplantation, resulting in lower efficiency.

In this study, we directly microinjected adenoviral CRISPR/Cas9 vectors into the dorsal aorta of chicken embryos to generate genetically modified chickens without PGC culture and modification in vitro. This presents a simple and effective method for the modification of chickens.

## 2. Results

### 2.1. Effective Editing Sites Screening in DF1 Cells

The *KRT75L4* gene was selected as the target gene because deletion in *KRT75L4* mediates the frizzle trait and knockout of the *KRT75L4* gene is expected to result in the frizzle trait, a modification visible in the phenotype where all contour feathers curl outward and upward [14]. To knockout all transcript variants of the chicken *KRT75L4* gene, sgRNAs targeting sequences in exons 1 and 2 were designed. To achieve high knockout efficiency, sgRNAs were integrated into the CRISPR/Cas9-expressed vector (pX459) and transfected into DF1 cells to evaluate their efficiency. After selection for *PuroR* expression using puromycin, PCR was performed with genomic DNA using the primer F1/R1 with barcodes for amplicon sequencing. Analysis using CRISPResso2 (http://crispresso.pinellolab.org/) software revealed sgRNA editing frequencies ranging from 31.15% to 77.90%. Notably, E2-sgRNA2 induced the highest mutation rates in *KRT75L4* (Figure 1A), with diverse modifications observed in alleles surrounding the cleavage site, including substitutions, insertions, and deletions, as depicted in Figure 1B.

E2-sgRNA2 was subsequently integrated into the commercially adenoviral shuttle vector containing the CRISPR/Cas9 system for producing the recombinant type 5 adenovirus (Figure 1D). AdV-EGFP and the AdV-CRISPR-EGFP were delivered into DF1 cells at an MOI of 5000 with polybrene (5 μg/mL) after being optimized (Appendix A). After 48 h of culture, AdV-EGFP produced cells for the control group (DF1_WT), while the AdV-CRISPR-EGFP system produced knockout cells (DF1_KO) (Figure 1C). Amplicon sequencing was performed to analyze editing efficiency, and the results showed a modification frequency of approximately 80%, including insertion (14.43%) and deletion (85.57%), with the most common mutation being an 8-bp deletion (27.38%) (Figure 1E,F).

### 2.2. Generation of Gamete Cells Editing Chimeric Chickens

Recombinant adenovirus type 5 containing the CRISPR/Cas9 vector was injected into the dorsal aorta of chicken embryos at stage 14–17HH after incubating for 60 h (Figure 2A). A total of 116 chicks (G0) were hatched from 213 injected chicken eggs, including 95 chicks injected with AdV-CRISPR-EGFP and 21 chicks injected with AdV-EGFP. Meanwhile, 97 embryos failed to hatch, including 84 embryos injected with AdV-CRISPR-EGFP and 13 embryos injected with AdV-EGFP. Parts of the chicks and adult chickens are shown in Figure 2B. Samples from the liver, heart, spleen, lung, kidney, stomach, intestine, brain, gonad, muscle, and skin from parts of the embryos that died after 17–19 days of incubation were collected and the modification efficiency of different tissues was detected by PCR and amplicon sequencing. The results are shown in Figure 2C, and the editing efficiency of the gonad was the highest (2.63–11.57%), while the brain had the lowest editing efficiency (1.58–2.86%). Further analysis of the editing patterns in gonads revealed different lengths of insertions (21.97%) and deletions (78.03%) occurring near the PAM sequence of sgRNA, with 8-bp deletions (4.21%) being the highest proportion. Moreover, significant modifications were observed in the liver (4.78–6.46%), skin (2.35–4.43%), spleen (2.34–10.58%), intestines (1.87–6.19%), chest muscle (2.60–5.34%), lung (2.39–7.13%), kidney (1.09–11.61%), heart (2.01–10.05%), and stomach (2.49–4.45%). Deletions constituted the highest proportion of all mutation types across tissues, with no substitutions observed. (Figure 2E). Additionally, 8-bp deletions were predominant among all gene modification results across tissues, consistent with the findings in DF1 cells (Appendix A). Blood collected from 27 chicks aged 1 to 2 months was used to detect the modification efficiency, and the results show that the efficiency was 0.79% ± 0.14%, demonstrating a significant difference (Figure 2D).

When these chicks were raised to sexual maturity, semen was collected from 23 roosters to detect the editing frequency and pattern. The results show that all the roosters injected with AdV-CRISPR-EGFP were genetically modified with a frequency ranging from 0.16% to 4.85%. Consistent with previous findings, the deletion of 8-bp was the most common gene modification (Figure 2F,G), indicating that these roosters injected with the AdV-CRISPR-EGFP vector were germline chimeras and potential founders for generating genetically modified offspring. We attempted the artificial insemination of wild-type Hy-Line hens with semen from these chimeras and analyzed the genotype of their offspring (G1). Unfortunately, we did not detect any heterozygous individuals among the nearly 1000 chicks.

## 3. Discussion

The present study reports an effective approach for CRISPR/Cas9-mediated targeted gene knockout in chicken germline cells without the use of in vitro culture for the genetic modification of PGCs. Instead of modifying PGCs in vitro, the recombinant adenovirus type 5 containing the optimized CRISPR/Cas9 vector was directly injected into the dorsal aorta of stage 14~17HH [15] embryos. This one-step approach successfully edited gamete cells, which subsequently colonized the embryonic gonad and produced genetically modified sperm after sexual maturity, as previously indicated in studies [16,17]. The modification efficiency of embryo gonads (2.63–11.57%) and sperm from chimeric roosters (0.16–4.85%) in the present study demonstrates that this method can facilitate the in vivo germline engineering of the chicken genome and production of genetically modified homozygous chickens through chimeras. Therefore, genes related to specific traits, such as chicken Na+/H+ exchange type 1 (*chNHE1*), associated with avian leukemia virus resistance [18], and myostatin (*MSTN*), involved in muscle growth [19], could potentially benefit from genetic modification using this method. Moreover, this approach could potentially be adapted for other avian species where the isolation, maintenance, and genetic modification of PGCs in cultures have not been successfully established, such as ducks, pigeons, and geese.

In addition, the adenovirus system may represent a more favorable delivery system for precise avian gene modification. Due to high efficiency in integrating transgenes into the genome [20,21], lentiviral vectors have been widely used for the generation of transgenic chickens [22,23,24,25]. However, they are lacking in biosafety and appropriacy for the generation of CRISPR/Cas9-mediated gene-modified chickens because the integration of the CRISPR/Cas9 and sgRNA cassettes can result in more off-target mutations and sustained modification [26,27]. Conversely, adenovirus type 5 has shown to be a more effective delivery system for CRISPR/Cas-mediated gene modification because of its high expression of adenovirus receptors and consequent high gene transduction efficiency [28]. Furthermore, replication-defective adenoviruses are genetically stable and do not integrate into the host genome [29]. The adenoviral CRISPR/Cas9 vector has been successfully used to modify genes such as *MSTN* in chicken leg muscles and *MLPH* in quail blastoderms via direct injection [12,30,31]. In the present study, all tissues, including germ cell-producing gonads, were genetically modified using the adenoviral CRISPR/Cas9 vector. The presence of mutated sperm from chimeric roosters further supports the one-step adenovirus system as a preferred delivery strategy for avian genetic engineering.

Adenoviral CRISPR/Cas9 vectors present a dual challenge due to the high immunogenicity of adenovirus proteins [32]. Adenoviruses can strongly stimulate the immune system by exposing pathogen-associated molecular patterns (PAMPs) on their outer shell, DNA, and intermediate parts. This stimulation can lead to the release of cytokines and chemokines, triggering excessive inflammation known as a cytokine storm [33]. In severe cases, this immune response can potentially induce risks, such as anaphylactic shock or even death [34,35], which may have significantly impacted the survival rate of G0 embryos (39%) in the present study. Additionally, keratins play crucial roles in the mechanical stability and integrity of epithelial cells and tissues, as well as in embryonic development [36,37]. Mutations in keratins can alter the micro-mechanical properties of the cytoskeleton and result in lethal neonatal phenotypes [38,39]. The significant knockout of *KRT75L4* in all tissues may exacerbate the reduced vitality of chicken embryos. Moreover, keratins regulate cell cycle progression, mitotic entry, and protein synthesis through interactions with key effectors [40,41]. Therefore, modifying *KRT75L4* in G0 embryos may disrupt fertilization and embryonic development, potentially resulting in the absence of G1 chicks.

## 4. Materials and Methods

### 4.1. Construction of Plasmids and Adenoviral Vectors

CRISPR/Cas9 constructs were prepared by cloning guide RNA sequences into the sgRNA scaffold of the pX459 vector (Addgene no. 48139) [42]. The oligonucleotides used for annealing the guide RNA sequences, sgRNA1-sgRNA8 (Appendix A), were designed for the first and second exons of *KRT75L4* (ENSGALG0001002413, GRCg7b) using the CHOPCHOP V3 [43]. They were selected based on high on-target specificity scores and synthesized by Sangon Biotech (Shanghai, China). Both oligonucleotides were phosphorylated, hybridized, and ligated into pX459 and cleaved by BbsI to generate pX459-KRT75L4-sgRNA. All plasmids were extracted using an Endo-Free Plasmid Mini Kit II (Omega Bio-Tek, Norcross, GA, USA). The recombinant adenovirus pAV [CRISPR]-hCas9: P2A: EGFP-U6>gRNA (abbreviated as AdV-CRISPR-EGFP, titer 1.15 × 10^12^ VP/mL) and pAV [Exp]-CMV>EGFP (abbreviated as AdV-EGFP, titer 1.79 × 10^12^ VP/mL) were packaged by Vector Builder (Guangzhou, China).

### 4.2. Cell Culture and Transfection

DF-1 cells were cultured in a DMEM/F12 medium (Gibco, Shanghai, China) supplemented with 10% FBS (Gibco, Shanghai, China) at 37 °C and 5% CO_2_. Transfections were performed when cells reached 70–80% confluency using 3.75 μL Lipofectamine 3000 and 5 μL P3000 reagents (Invitrogen, Carlsbad, CA, USA) with 2.5 μg of pX459-*KRT75L4*-sgRNA plasmid DNA. Cells were cultured for one week, and stable cells expressing *PuroR* were selected using puromycin. AdV-CRISPR-EGFP and AdV-EGFP adenovirus were gently added to DF-1 cells cultured in well plates at 60–80% confluency. Polybrene was added during transfection. After incubation, cells were washed twice with phosphate-buffered saline (PBS), and a fresh culture medium was added.

### 4.3. Microinjection of Adenoviral Vector

Hy-Line white fertilized eggs were sanitized using 75% ethanol after 60 h of incubation (Brinsea, Ova-Easy Advance Series II, Weston-super-Mare, UK). A small opening less than 1 cm in diameter was made at the blunt end of each recipient egg using fine-tip tweezers to provide access to the dorsal aorta of stage 14–17HH embryos. In total, 2 μL of AdV-CRISPR-EGFP was injected into the left dorsal aorta using a pulled glass micropipette, while AdV-EGFP was used as a control. The opening in the egg was sealed with paraffin film, and the eggs were positioned with the pointed end down for 8 days at 37.8 °C and 60% relative humidity before being transferred to an automatic egg incubator.

### 4.4. Animal Breeding and Generation

Eggs injected with adenoviral vectors were incubated at 37.8 °C and 60% relative humidity in a digital cabinet egg incubator. All chickens were reared in the Animal Husbandry and Veterinary Research Institute (Shanghai Academy of Agricultural Science, Shanghai, China) under standard conditions (16 h light/8 h dark cycle and ad libitum access to food and water). Male G0 chicks were raised to sexual maturity, and semen was collected and examined for the presence of the *KRT75L4*-KO transgene. GO roosters (chimeric male chicks raised to sexual maturity) with the highest efficiency of *KRT75L4* knockout were mated with wild-type females, and offspring (G1) were screened for *KRT75L4* knockout. Experimental protocols, including animal care and experimental activities, were approved by the Animal Ethics Committee at Shanghai Jiao Tong University in China, approval No. 202206005 (21 June 2022).

### 4.5. Analysis of Editing Efficiency

Genomic DNA from cells and tissues was extracted using the Tissue DNA Kit (Omega Bio-Tek, Norcross, GA, USA). PCR amplification to detect editing efficiency was performed using the primers E1/2-sgRNA1~4-F/R and Novaseq-F/R (Appendix A) with the 2× Phanta Master Mix (Vazyme, Nanjing, China). The PCR products were pooled and sequenced with 250 bp paired-end (PE250) reads on a NovaSeq instrument by Novogene (Beijing, China). PCR amplification to identify G1 was performed using G1-F/R primers (Appendix A). Data were analyzed using CRISPResso2 software to calculate gene editing efficiency and mutation patterns.

## 5. Conclusions

The genetic modification of chickens requires the culture and modification of PGCs in vitro; however, it is technically difficult and may compete with endogenous PGCs. The one-step genetic modification approach, achieved by the direct microinjection of an adenoviral CRISPR/Cas9 vector into the dorsal aorta of chicken embryos, offers a simple and effective means to generate modifications in chicken germ cells in vivo without the need for PGC culture and modification. This method can be readily applied to avian species lacking established PGC culture procedures, facilitating genetic modification studies in poultry breeding for disease resistance and growth efficiency.

## Figures and Tables

**Figure 1 ijms-25-08692-f001:**
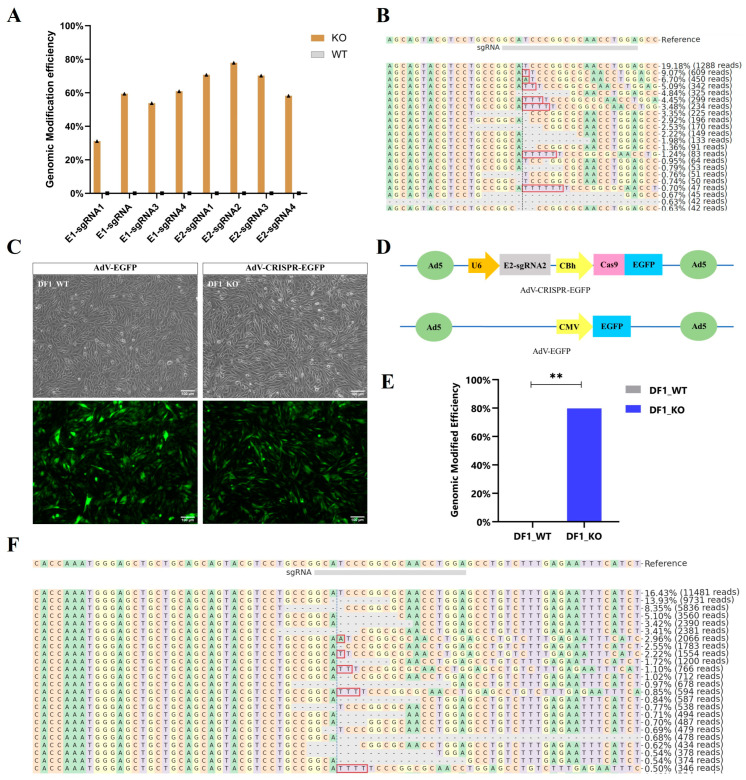
Modification efficiency and patterns in DF1 cells. (**A**) The modification efficiency of pX459-sgRNAs plasmids. (**B**) Visualization of the distribution of identified alleles around the cleavage site for the pX459-E2-sgRNA2 in DF1 cells. Nucleotides are indicated by unique colors (A = green; C = red; G = yellow; T = purple). Substitutions are shown in bold font. Red rectangles highlight inserted sequences. Horizontal dashed lines indicate deleted sequences. The vertical dashed line indicates the predicted cleavage site. (**C**) Phase contrast (upper) and fluorescence (lower) images of DF-1 cells transfected with the AdV-EGFP and AdV-CRISPR-EGFP virus for 48 h. (**D**) Schematic of the AdV-CRISPR-EGFP and AdV-EGFP vector. (**E**) Modification efficiency of the adenoviral CRISPR/Cas9 vector in DF1 cells, ** *p* < 0.01. (**F**) Visualization of the distribution of identified alleles around the cleavage site for AdV-CRISPR-EGFP in DF1 cells.

**Figure 2 ijms-25-08692-f002:**
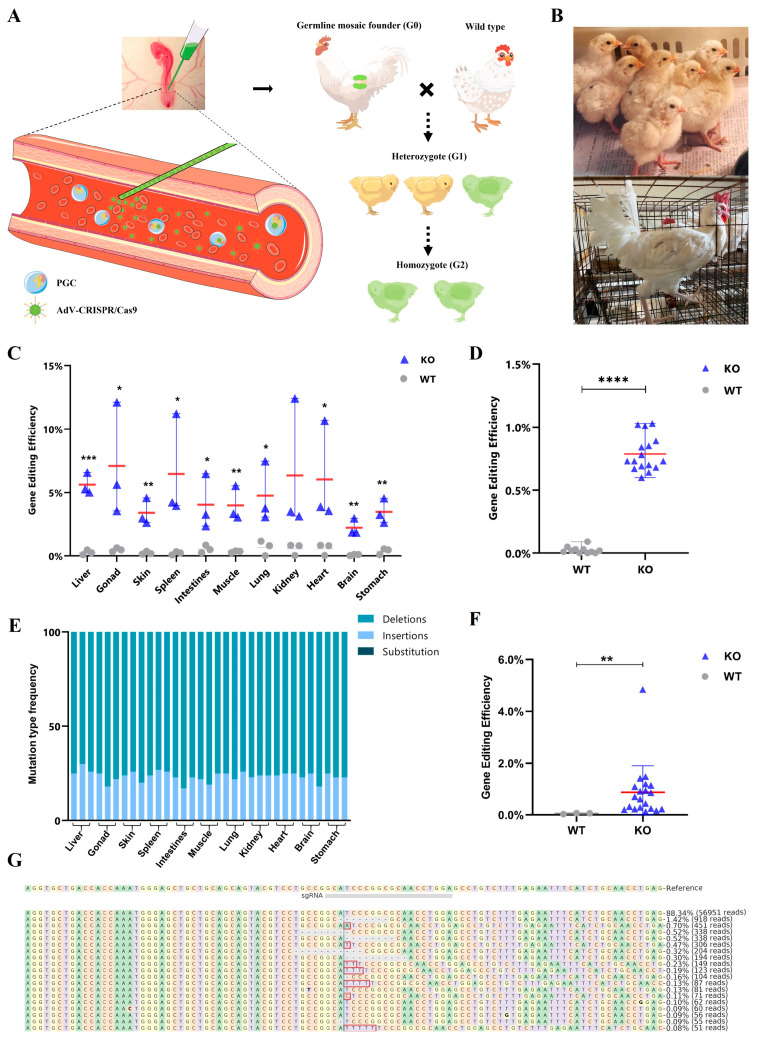
Modification in chicken embryos. (**A**) Schematic diagram of dorsal aorta microinjection of adenovirus for germ cell editing. (**B**) Representative images of chicks and roosters injected with AdV-CRISPR-GFP. (**C**) Modification efficiency of adenoviral CRISPR/Cas9 vector in tissues of embryos. Data are presented as mean ± SD, n = 3, * *p* < 0.05, ** *p* < 0.01, and *** *p* < 0.001. (**D**) Blood modification efficiency of chicks. Data are presented as mean ± SD; 16 chicks were injected with AdV-CRISPR-EGFP vector, and 11 chicks were injected with AdV-EGFP vector, **** *p* < 0.0001. (**E**) Frequency distribution of sequence modifications classified as insertions, deletions, and substitutions in tissues of embryos. (**F**) Semen modification efficiency of roosters. Data are presented as mean ± SD; 23 roosters were injected with AdV-CRISPR-EGFP vector, and 3 roosters were injected with AdV-EGFP vector, ** *p* < 0.01. (**G**) Visualization of distribution of identified alleles around cleavage site for AdV-CRISPR-EGFP in semen.

## Data Availability

Data is contained within the article and Appendix A.

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
