# Peer review of "One-Step Genetic Modification by Embryonic Doral Aorta Injection of Adenoviral CRISPR/Cas9 Vector in Chicken"

_ijms, 2024, doi:10.3390/ijms25168692_

Round 1
Reviewer 1 Report
Comments and Suggestions for Authors
The authors describe here the generation of mosaic deletions in chickens after administering adenoviral CRISPR deletion of a keratin gene by injecting the adenoviral construct into the embryonic dorsal aorta. They could demonstrate a relatively low percentage of mutations in several organs. There was a high number of non-hatchers. After 2 months, hardly any traces of the manipulation were detected in the blood. The study raises a number of questions regarding the efficacy of the approach. However, more importantly, there are safety issues to be observed if this methods is going to be used for the production of better meat quality or advantages in animal survival and husbandry. First of all, the authors should provide the official registration number of the animal experiment permission. Secondly, they should make transparent that they plan to genetically modify poultry destined for human food.
Comments on the Quality of English Language
Language errors appear throughout the manuscript. They should explain "G0 rooster."
Author Response
The authors describe here the generation of mosaic deletions in chickens after administering adenoviral CRISPR deletion of a keratin gene by injecting the adenoviral construct into the embryonic dorsal aorta. They could demonstrate a relatively low percentage of mutations in several organs. There was a high number of non-hatchers. After 2 months, hardly any traces of the manipulation were detected in the blood. The study raises a number of questions regarding the efficacy of the approach. However, more importantly, there are safety issues to be observed if this method is going to be used for the production of better meat quality or advantages in animal survival and husbandry. First of all, the authors should provide the official registration number of the animal experiment permission.
Response:
Thank you for your valuable feedback. All the experimental protocols including animal care and experimental activities were approved by the Animal Ethics Committee at Shanghai Jiao Tong University in China and the relevant official registration number has been added in line 228 in revised manuscript.
Secondly, they should make transparent that they plan to genetically modify poultry destined for human food.
Response:
In the present study, we do not plan to genetically modify poultry as food for human, and there is currently no genetically edited poultry for human food. Besides, compared to the PGCs-mediated methods, the method in this study also is unlikely to have safety issues because the adenoviral CRISPR/Cas9 vector is genetically stable and do not integrate, which has been descripted in line 153-162 in the revised manuscript.
Language errors appear throughout the manuscript. They should explain "G0 rooster."
Response:
In our revised version, we have meticulously proofread revised the manuscript to rectify any linguistic inaccuracies and ensure the manuscript meets high standards of language proficiency, and the all changed places are highlighted in yellow in the manuscript.
"G0 roosters" is chimeric male chicks raised to sexual maturity which may be potential founders for generating genetically modified offspring. In the revised manuscript, we have provided explanation of "G0 rooster" in line223-224.
Reviewer 2 Report
Comments and Suggestions for Authors
This paper presents a novel approach for genetic modifications in poultry through an embryonic doral aorta vector injection. This is an innovative and interesting strategy; however, the authors could provide more details to the potential use of the CRISPR/Cas9 technology in improving que quality traits in avian species: what qualities could be improved? Are there any other similar techniques that have been already tested out? If so, what advantages could this novel technique bring along?
The methodology is well-described and supported with comprehensive figures, however I am interested in the semen collection and evaluation: how were the samples collected? Was the overall semen quality evaluated, and if so, how? How was the presence of the transcript assessed?
In the Discussion section, the authors talk about a possible complication resulting from the high immunogenicity of the protein. How could be the strong immunogenic reaction ameliorated or trespassed?
The authors could also discuss any limitations that may have affected the outcomes of the study.
Author Response
This paper presents a novel approach for genetic modifications in poultry through an embryonic doral aorta vector injection. This is an innovative and interesting strategy; however, the authors could provide more details to the potential use of the CRISPR/Cas9 technology in improving que quality traits in avian species: what qualities could be improved?
Response:
Thank you for your insightful comments and suggestions. In our revised manuscript, we expand upon the discussion regarding the specific quality traits that could be targeted for improvement using CRISPR/Cas9 in line145-148. This included detailing traits such as disease resistance (chNHE1) and growth efficiency (MSTN) that could benefit from genetic modifications.
Are there any other similar techniques that have been already tested out? If so, what advantages could this novel technique bring along?
Response:
There are some similar techniques that have been already tested out (Chen et al., 2016, Challagulla A et al.,2020) in which they injected lentiviral vector or plasmid with lipo2000 into the embryos. We have provided a comparative analysis with other similar techniques that have been tested in avian species in line 152-167. This discussion will highlight the advantages and unique aspects of our novel technique, particularly focusing on its potential to achieve targeted and efficient genetic modifications, which may offer advantages such as scalability, and applicability across different avian breeds.
- Chen P R, Shin S, Choi Y M, et al. Overexpression of G0/G1 Switch Gene 2 in Adipose Tissue of Transgenic Quail Inhibits Lipolysis Associated with Egg Laying [J]. International Journal of Molecular Sciences, 2016, 17(3): 384.
- Challagulla A, Jenkins KA, O'Neil TE, et al. Germline engineering of the chicken genome using CRISPR/Cas9 by in vivo transfection of PGCs. Anim Biotechnol. 2023;34(4):775-784.
The methodology is well-described and supported with comprehensive figures, however I am interested in the semen collection and evaluation: how were the samples collected?
Response:
When the chimeric roosters reach approximately five months of age, skilled breeders begin training them for ejaculation until the desired semen was collected in eppendorf tubes and immediately transfer to lab in the low temperature, and we proceed immediately to the next step.
Was the overall semen quality evaluated, and if so, how?
Response:
Regarding semen quality evaluation, we just simply evaluated the concentration and motility under a microscope so that it can be used for genetic testing and artificial insemination.
How was the presence of the transcript assessed?
Response:
We do not analyze the transcript of the semen from chimeric roosters in the study because we are more concerned about whether the sperm are genetically modified. In the subsequent work if we get the heterozygous or homozygous editing individual, then analysis of transcript is necessary.
In the Discussion section, the authors talk about a possible complication resulting from the high immunogenicity of the protein. How could be the strong immunogenic reaction ameliorated or trespassed?
Response:
We think the adeno-associated virus (AAV) may be a better strategy to mitigate or overcome strong immunogenic reactions. Compared to adenovirus, AAV offers higher titers, which can further enhance gene editing efficiency. Moreover, AAV exhibits milder pathogenicity and immunogenicity in humans compared to adenovirus vectors, making it safer. Importantly, AAV vectors do not introduce additional mutations during the editing process (Bijlani Swati et al., 2022). However, due to the limited packaging capacity of AAV vectors, excluding viral inverted repeat sequences, they can only accommodate approximately 4.5 kb, which restricts the size of the Cas9 gene, as well as the design of expression sequences and regulatory elements.
- Bijlani S, Pang KM, Sivanandam V, Singh A, Chatterjee S. The Role of Recombinant AAV in Precise Genome Editing. Front Genome Ed. 2022;3:799722.
The authors could also discuss any limitations that may have affected the outcomes of the study.
Response:
In our revised manuscript, we have discussed the limitations encountered during the course of our research in line 168-183, such as adenoviruses and deletion of KRT75L4 may strongly affect the outcomes of the study.